# Disproportionate Effect of Sub-Micron Topography on Osteoconductive Capability of Titanium

**DOI:** 10.3390/ijms20164027

**Published:** 2019-08-18

**Authors:** Juri Saruta, Nobuaki Sato, Manabu Ishijima, Takahisa Okubo, Makoto Hirota, Takahiro Ogawa

**Affiliations:** Weintraub Center for Reconstructive Biotechnology, Division of Advanced Prosthodontics, UCLA School of Dentistry, Los Angeles, CA 90095-1668, USA

**Keywords:** acid-etching, micro-rough, bone regeneration, sub-micro-rough, bone integration, osseointegration, dental implants, orthopedic implants

## Abstract

Titanium micro-scale topography offers excellent osteoconductivity and bone–implant integration. However, the biological effects of sub-micron topography are unknown. We compared osteoblastic phenotypes and in vivo bone and implant integration abilities between titanium surfaces with micro- (1–5 µm) and sub-micro-scale (0.1–0.5 µm) compartmental structures and machined titanium. The calculated average roughness was 12.5 ± 0.65, 123 ± 6.15, and 24 ± 1.2 nm for machined, micro-rough, and sub-micro-rough surfaces, respectively. In culture studies using bone marrow-derived osteoblasts, the micro-rough surface showed the lowest proliferation and fewest cells attaching during the initial stage. Calcium deposition and expression of osteoblastic genes were highest on the sub-micro-rough surface. The bone–implant integration in the Sprague–Dawley male rat femur model was the strongest on the micro-rough surface. Thus, the biological effects of titanium surfaces are not necessarily proportional to the degree of roughness in osteoblastic cultures or in vivo. Sub-micro-rough titanium ameliorates the disadvantage of micro-rough titanium by restoring cell attachment and proliferation. However, bone integration and the ability to retain cells are compromised due to its lower interfacial mechanical locking. This is the first report on sub-micron topography on a titanium surface promoting osteoblast function with minimal osseointegration.

## 1. Introduction

Titanium and titanium alloy have been widely used in the fields of orthopedic surgery and dentistry owing to their excellent mechanical properties, high corrosion resistance, and suitable biocompatibility [1,2,3]. Uses include endosseous implants and various bone regenerative devices, such as pins and screws to immobilize bone and plates and scaffolds to guide bone generation [1,4,5,6,7,8]. To improve the biocompatibility of titanium, particularly its osteoconductivity, various methods of surface modification have been developed to roughen titanium surfaces. Surface modifications consist of mechanical, chemical, and physicochemical treatments, as well as other coating-based methods, including machining, sand-blasting, acid-etching, anodization, plasma spraying, laser treatment, apatite-coating, or a combination of these [9,10].

Surface topography and roughness influence the biological responses of osteoblast cells [11]. For improved osseointegration of implants, researchers have assessed the impact of surface roughness at the micro-scale [12]. Surface roughness not only increases the surface area but also triggers biological changes, such as the skeletal and morphological alteration of cells. These changes can affect planar cell polarity, as well as gene expression and the differentiation and maturation of osteoblasts [13,14]. In the past, studies have reported that, even in the absence of additional osteogenic factors, micro-textured titanium surfaces can promote the differentiation and maturation of osteoblasts and implant osseointegration better than relatively smoother surfaces, such as machined surfaces [15,16]. In fact, bone formation occurs in structurally complex areas generated through the process of bone deposition and resorption by osteoblasts and osteoclasts [12,14], and these areas present micro-scale roughness features. Titanium surfaces with various micro-scale topographies developed to date have been shown to provide osteoblasts with structural mimetics and thereby biological cues to promote new bone formation [17].

Presently, a micro-rough titanium surface created by acid-etching is one of the most commonly used surfaces in endosseous implants. This particular surface is characterized by a compartmental structure consisting of pits and peaks, ranging in size from 1 to 5 µm [18]. Sulfuric acid (H_2_SO_4_) is used in this surface modification, although many other strong acids, such as hydrochloric acid (HCl), nitric acid (HNO_3_), and hydrogen fluoride (HF), have been tested [1,19]. The acid-etched micro-rough titanium surface promotes osteoblastic differentiation, and thereby bone formation around it, better than smoother surfaces, such as a machined surface [1,20]. The improved osteoconductivity provides the therapeutic benefits of faster bone formation and, more importantly, a stronger anchorage of implants into bone, referred to as bone–implant integration or osseointegration. However, sub-micron topographies have rarely been developed or applied to titanium implants, and their biological impacts are largely unknown.

We have created a new titanium surface with sub-micro-topography. The purpose of this study was to examine the rate of osteoblastic differentiation on this new surface as well as its interfacial locking ability at the cellular and tissue levels.

## 2. Results

### 2.1. Surface Morphology of Titanium

Macroscopically, the machined surface was light gray with a metallic luster (Figure 1a,d), whereas the micro-rough surface was dark gray with no metallic luster (Figure 1b,e). The sub-micro-rough surface was grayish, similar to the machined surface, but with less metallic luster (Figure 1c,f).

Low-magnification scanning electron microscopy (SEM) images of the machined surface showed parallel traces formed during the concentric machining process (Figure 1g,j). High-magnification images of the machined surface showed undefined irregularities (Figure 1m). The micro-rough surface showed a typical micro-roughened morphology, consisting of microscale pits with 1–5 µm in peak-to-peak distance (approximately 1.5 µm on average) (Figure 1h,k,n). Low-magnification images of the sub-micro-rough surface showed no recognizable roughness with faintly machine traces (Figure 1i). Higher magnification images of the sub-micro-rough surface showed roughness consisting of pits smaller than those on the micro-rough surface (Figure 1l,o). The pits were 0.1–0.5 µm, with an average size of 0.15 µm (Figure 1o).

### 2.2. Quantitative Topographical Evaluations of Titanium Surfaces

To identify potential measurable differences in surface morphology among the three surfaces, quantitative assessments of 3-dimensional profiles were performed. The results showed that roughness parameters such as average roughness and mean peak-to-valley height were significantly greater on the sub-micro-rough surface than on the machined surface and significantly greater on the micro-rough surface than on the sub-micro-rough surface (Figure 2a,b). The average width of roughness profile elements was significantly higher on both the machined and micro-rough surfaces than on the sub-micro-rough surface, whereas there was no significant difference between the machined and micro-rough surfaces (Figure 2c). The skewness of the roughness profile was highest on the micro-rough surface and lowest on the sub-micro-rough surface (Figure 2d).

### 2.3. Cell Attachment, Proliferation, and Functional Phenotypes

The effect of cell attachment onto titanium surfaces was assessed based on the numbers of cells attached to the titanium surfaces after a 24 h culture period. The number of osteoblasts attached to each titanium disk was evaluated after 24 h of incubation using a WST-1-based colorimetric assay (Figure 3a). After 24 h of seeding, significantly more cells were attached to the machined and sub-micro-rough surfaces than the micro-rough surface. The number of attached cells was 2.8 times higher for the machined surface and 2.4 times higher for sub-micro-rough surface than for the micro-rough surface (Figure 3a). 

The number of propagated cells on each titanium surface was assessed based on the cell density on the titanium surfaces on day 3 of the culture period. Cell density, as measured on day 3, was significantly greater on the machined surface than on the micro-rough or sub-micro-rough surfaces and was significantly greater on the sub-micro-rough surface than on the micro-rough surface. The cell density was 5.3 times greater for the machined surface and 2.1 times greater for the sub-micro-rough surface than for the micro-rough surface (Figure 3b). 

To assess osteoblastic phenotypes on titanium surfaces, bone-related gene analysis was performed using quantitative PCR at two time points. At day 7 of culture, quantitative PCR showed that the expression of collagen type I alpha 1 chain (*Col1a1*), an early marker of osteoblastic differentiation, was significantly greater on the micro-rough surface than on the machined surface, and greater still on the sub-micro-rough surface. Expression was upregulated 1.4- and 3.1-fold on the micro-rough and sub-micro-rough surface, respectively, compared to that on the machined surface. At day 14 of culture, the trends seen on day 7 persisted, with the sub-micro-rough surface showing the highest expression of *Col1a1* and the machined surface showing the lowest (Figure 3c). 

At day 7 of culture, the expression of osteopontin (*Opn*), a mid- to late-stage osteoblastic marker, was upregulated on the micro-rough and sub-micro-rough surfaces. In particular, marked upregulation was observed on the sub-micro-rough surface. At day 14 of culture, there was also a significant increase in *Opn* expression on both micro-rough and sub-micro-rough surfaces compared to levels on the machined surface, although there was no significant difference between the micro-rough and sub-micro-rough surfaces (Figure 3d). 

The level of calcium deposition was evaluated as a later-stage osteoblastic phenotype. Total calcium deposition on day 14 was significantly greater on the micro-rough surface than on the machined surface and was greater still on the sub-micro-rough surface (Figure 3e, bottom). The Alizarin red-positive area was largest on the sub-micro-rough surface and smallest on the machined surface, confirming the calcium deposition results (Figure 3e, top).

### 2.4. Detachment of Osteoblasts from Different Titanium Surfaces

The detachment test was performed to determine the cell–titanium surface attachment strength in vitro. The detachment test performed on culture day 3 showed that approximately 80% of cells detached from the machined and sub-micro-rough surfaces, whereas only 25% of cells detached from the micro-rough surface (Figure 4a). There was no significant difference between the machined and sub-micro-rough surfaces. Detachment tests on days 7 and 14 showed a similar trend to that on day 3. The percentage of detached cells was significantly reduced on the micro-rough surface but not on the sub-micro-rough surface compared to that on the machined surface (Figure 4a). 

SEM qualitative observation was performed to confirm the results of the detachment tests. On day 3, most cells had detached from the machined and sub-micro-rough surfaces, whereas a higher percentage of cells remained present on the micro-rough surface (Figure 4b–g). On day 14, a greater number of cells were present on titanium disks than on day 3. A majority of the cells detached from the machined and sub-micro-rough surfaces after the detachment test, whereas a significant number of cells remained adhered to the micro-rough surface (Figure 4h–m).

### 2.5. Biomechanical Strength of Bone and Titanium Integration

A push-in value that directly represents the shear strength of integration between the bone and implant was used to evaluate the osseointegration. The in vivo strength of bone–implant integration as evaluated by the push-in test was 5.5 times higher for the micro-rough surface than for the machined surface. The sub-micro-rough surface did not show a significant increase compared to the value of the machined surface (Figure 5). The strength of bone–implant integration for the sub-micro-rough surface was significantly lower than that for the micro-rough surface.

## 3. Discussion

In this study, we aimed to evaluate the osteoconductivity and anchorage capability of a sub-micro-rough titanium surface as compared to machined and micro-rough surfaces. All in vitro results indicated that the sub-micro-rough surface promoted osteoblastic differentiation even more than the micro-rough surface. Interestingly, this enhanced osteoblastic differentiation did not result in an increase in in vivo bone and implant integration, probably due to the limited mechanical retention of bone tissue. Performance of the detachment assay on cultured osteoblasts revealed that the ability to retain cells was substantially reduced on the sub-micro-rough surface compared with that of the micro-rough surface. Despite the effective surface roughening provided by the low-temperature acid-etching, the cell retention ability remained similar to that of the machined surface. Together, our results indicate that the sub-micro-rough surface exhibits new surface features that effectively promote osteoblastic differentiation but not the mechanical interfacial engagement of cells and bone tissue. The fact that the surface roughness has disproportionate effects on osteoconductivity on the one hand and on bone and titanium integration on the other represents a significant finding.

Although the effect of nano-features on titanium surfaces has been studied, little information was available on the effect of sub-micro-features. We developed a new method for creating smaller scale compartmental structures, ranging from 0.1 to 0.5 µm, on titanium than is typically seen on acid-etched micro-rough titanium. The formation of compartments was even and uniform under the controlled temperature. Given the unique biological effects demonstrated in this study, this surface may be useful when an increased osteoconductivity but not a strong anchorage in bone is desired, such as in temporal bone regeneration devices. For instance, titanium mesh plates, pins, and screws are used for scaffolding and guiding new bone formation and fixing bone pieces, with the intention of removal after the purpose has been accomplished. Because of the considerable increase in the osteoconductivity of this sub-micro-rough surface, bone healing around the device would be promoted. In contrast, the minimal increase in bone–titanium integration ability would facilitate relatively easy removal of the device.

We examined the cell attachment, proliferation, and functional phenotypes of osteoblast-like rat bone marrow in this study. The rat bone marrow cells used in this experiment are known to differentiate towards an osteoblast-like phenotype when supplemented with dexamethasone and β-glycerophosphate [21,22,23,24]. In addition, many studies related to osteoblasts have been reported using various cells, such as bone-marrow-derived mesenchymal stem cells, MC3T3-E1, and MG-63 [25,26,27,28,29,30,31]. Osteoblast differentiation methods have been reported to use many kinds of reagents, including 1,25-dihydroxyvitamin D_3_ [32], hormones [33,34], growth factors [35,36], bone morphogenetic proteins [37,38], aluminum chloride (AlCl_3_) [39], sodium fluoride (NaF) [35], prostaglandins [40,41], β-glycerophosphate [30], and ascorbic acid [31]. There are also many reports on the timing of administering differentiation-inducing reagents to these cells [27,42,43]. Cells harvested from rat bone marrow were differentiated in this study using an osteogenic induction medium containing ascorbic acid, β-glycerophosphate, and dexamethasone. Since titanium materials do not have osteogenic induction capability in the material itself, it was necessary to differentiate and induce cells before seeding in order to examine osteoconduction on each type of titanium surface in the present study. In past studies, we have reported many changes in the differentiation, proliferation, and genetic phenotype of mature osteoblast cells before seeding cells using the osteogenic induction medium [44,45,46,47]. It is necessary to use various osteogenic induction reagents, change the timing of the addition of the osteogenic induction medium, and conduct experiments in a comprehensive manner in future studies. Moreover, since cells harvested from bone marrow cells did not use cell-sorting in this study, we should also conduct detailed studies on sub-micro-rough surfaces using rat and human mesenchymal stem cells.

Another important advantage provided by this sub-micro-rough surface is the amelioration of the adverse effects of micro-rough titanium. Micro-rough titanium surfaces are known to decrease the number of cells that attach during the initial stage of cell culture as well as the rate of cell proliferation [45,48]. This was observed in the present study, consistent with previous reports. The sub-micro-rough surface increased the number of attached cells and the cell density more than 2-fold compared with those of the micro-rough titanium. In particular, the number of attached cells on the sub-micro-rough surface was equivalent to that on the machined surface. The amount of bone formed is dependent on the number of osteoblasts present at the titanium surface. This new biological effect of the sub-micro-rough surface is meaningful from this perspective.

Another important finding from the present in vitro study is that the sub-micro-rough surface challenges a known principle of osteoblasts, i.e., the inverted relationship between proliferation and differentiation. For instance, the rate of proliferation is reduced on rougher biomaterial surfaces, although rougher surfaces have an advantage in that they promote cellular differentiation [45,49]. Similarly, micro-roughened titanium surfaces promote osteoblastic differentiation better than machined, smooth surfaces and result in faster bone formation. However, bone volume is ultimately reduced compared to that achieved with a machined surface owing to the reduction in osteoblastic proliferation. Consistent with this, several in vitro studies have shown that cell density and proliferative activity are reduced on micro-rough titanium surfaces compared with those on relatively smoother surfaces [50,51]. Given this, we anticipated that the sub-micro-rough surface would show a higher cell density than the micro-rough surface, which turned out to be true. However, the sub-micro-rough surface also showed accelerated osteoblastic differentiation, as demonstrated by the upregulated expression of osteoblastic genes. This was supported by the increased deposition of calcium on the sub-micro-rough surface. However, since we are using an osteogenic induction medium in these experiments, it is possible that osteoblasts are somewhat emphasized in differentiation rather than proliferation. Since titanium materials do not have osteoinduction capabilities, the cells require an osteogenic induction medium. Therefore, we need to further investigate the effects of the surfaces by creating a concentration gradient in the osteogenic induction medium and reducing the reagents contained in it.

We conducted an in vivo push-in test to evaluate the ability of implants to integrate with bone. Bone–implant integration was the strongest for the micro-rough surface, and there was no significant increase from the use of a sub-micro-rough surface. Given the strong promotion of osteoblastic differentiation and elevation of calcium deposition on the sub-micro-rough surface, we hypothesize that the surface pits of the low-temperature acid-etched titanium, which were 0.1 to 0.5 µm in width and 0.1 µm in depth, were not large or deep enough for the extracellular matrix to effectively engage.

In the present study, we did not histologically evaluate the bone–implant interface of the three kinds of implant after the push-in test. We placed acid-etched implants and machined implants in a rat femur model in a previous study and examined their bone histological imaging around implants with Goldner’s trichrome stain [45]. From these results, the bone–implant contact was significantly higher on the acid-etched surface than on the machined surface. Furthermore, we also reported the surface of the acid-etched implant and machined implant after push-in tests with implant surface morphology based on SEM and elemental composition analysis of the implant surface using energy dispersive X-ray spectroscopy (EDS) [52]. The machined implant after a push-in test at 4 weeks showed exposure without notable bone-like structures, with the accompanying EDS spectrum representing clear and high peaks of titanium and lower peaks of phosphorous and calcium. On the other hand, acid-etched implants showed the robust formation of bone-like structures, which were represented on the EDS spectrum by clear peaks of calcium and phosphorus. Moreover, the sub-micro-rough surface may have a bone–implant contact equivalent to that of the machined surface because there was no significant difference between them in terms of push-in value after 2 weeks of implantation. Considering the above, it may be predicted that many bone-like structures are detached from the implant interface on the sub-micro-rough surface, as on the machined surface. Therefore, we will need to conduct morphological and histological evaluations on the sub-micro-rough surface between implants and bone tissue after push-in tests using histological analysis and SEM in future studies. We should also evaluate the elemental composition of the sub-micro-rough implant surface using EDS.

This study revealed for the first time a disproportionate effect of titanium surface roughness on osteoblastic differentiation and bone integration capability through the introduction of a newly developed sub-micro-rough surface. This sub-micro-rough titanium is unique in terms of promoting osteoblastic differentiation while increasing proliferation. The present study also provides a new strategy for designing titanium surfaces with better osteoconductivity and minimal interlocking to bone tissue.

## 4. Materials and Methods

### 4.1. Titanium Sample Preparation

Disks (20 mm in diameter and 1 mm in thickness) and cylindrical implants (1 mm in diameter, 2 mm in length) of grade 2 commercially pure titanium were prepared with a machined surface (Figure 1a,d), a micro-rough surface introduced by regular acid-etching with 67% (*w*/*w*) H_2_SO_4_ (Sigma-Aldrich, St. Louis, MO, USA) at 120 °C for 75 s (Figure 1b,e), or a sub-micro-rough surface introduced by low-temperature acid-etching with 67% H_2_SO_4_ at 95 °C for 75 s (Figure 1c,f). These were placed in a sealed container and stored in a dark room (temperature, 23 °C; humidity, 60%) for 4 weeks.

### 4.2. Titanium Surface Characterization

The surface morphologies of the machined, micro-rough, and sub-micro-rough surfaces were examined by SEM (Nova 230 Nano SEM, FEI, Hillsboro, OR, USA). The surface roughness was quantified by measuring average roughness (*Ra*), mean peak-to-valley height (*Rz*), average width of roughness profile elements (*Rsm*), and skewness of roughness profile (*Rsk*) values using a 3-dimensional profiler (Mex, Alicona Imaging GmbH, Raaba, Graz, Austria).

### 4.3. Osteoblastic Cell Culture

Bone marrow-derived osteoblastic cells were isolated from the femurs of 8-week-old male Sprague–Dawley rats and placed into alpha-modified Eagle’s medium supplemented with 15% fetal bovine serum, 50 mg/mL ascorbic acid, 10 mM Na-β-glycerophosphate, 10^−8^ M dexamethasone, and antibiotic–antimycotic solution containing 10,000 units/mL penicillin G sodium, 10,000 mg/mL streptomycin sulfate, and 25 mg/mL amphotericin B, as previously described [53]. Cells were incubated in a humidified atmosphere of 95% air and 5% CO_2_ at 37 °C. At 80% confluency, cells were detached using 0.25% trypsin-1 mM EDTA-4Na and seeded onto titanium disks placed in a 12-well culture dish at a density of 3 × 10^4^ cells/cm^2^. The culture medium was renewed every 3 days.

### 4.4. Cell Attachment and Density Assay

Initial attachment of cells was evaluated by measuring the number of cells attached to titanium disks after 24 h of incubation. Propagated cells were also quantified as cell density on day 3 of culture. These quantifications were performed using a WST-1-based colorimetric assay (WST-1, Roche Applied Science, Mannheim, Germany). A culture well was incubated at 37 °C for 1 h with 100 µL of WST-1 reagent. The amount of formazan product was measured using a multi-detection microplate reader (Synergy^TM^ HT; BioTek Instruments, Inc., Winooski, VT, USA) at a wavelength of 450 nm.

### 4.5. Gene Expression Analysis

Total RNA was extracted from cultures on days 7 and 14 using the TRIzol reagent (Life Technologies, Carlsbad, CA, USA) and purified using Direct-zol^TM^ RNA MiniPrep (ZYMO RESEARCH Co., Irvine, CA, USA) according to the manufacturer’s protocol when the cells had reached 70% confluence. The total amount of RNA and its purity were determined using a NanoDrop^TM^ One (Thermo Fisher Scientific, Waltham, MA, USA) spectrophotometer. RNA with A260/A280 ratios between 1.8 and 2.0 were considered to be of high purity. Purified RNA was stored at –80 °C. Purified RNA was treated with the DNase I reagent (Invitrogen, Carlsbad, CA, USA) and converted into complementary DNA (cDNA) using a SuperScript^®^ VILO^TM^ cDNA System Kit (Invitrogen) as outlined by the manufacturer. Total RNA (1 µg) was used, and reverse transcription was performed in a final volume of 20 µL, following the manufacturer’s instructions. cDNA samples were stored at –20 °C for later use. Real-time PCR was conducted with the QuantStudio^TM^ 3 Real-Time PCR System (Applied Biosystems, Waltham, MA, USA) using TaqMan^®^ universal master mix II (Applied Biosystems). Relative gene expression of the following genes was quantified: *Col1a1* (Assay ID: Rn01463848_m1, Applied Biosystems) and *Opn* (Assay ID: Rn0068103_m1, Applied Biosystems). Glyceraldehyde-3-phosphate dehydrogenase (*Gapdh*) (Assay ID: Rn01775763_g1, Applied Biosystems) was used as a “housekeeping” gene for normalizing mRNA levels. Quantification of each gene was performed in triplicate, using 96-well plates and the cDNA obtained as described previously [54]. TaqMan gene expression assays specific to each gene were performed under conditions recommended by the manufacturer. Analysis of relative gene expression was performed with the 2^–ΔΔCt^ method [55]. Thermocycling conditions were as follows: 95 °C for 10 min, followed by 50 cycles at 95 °C for 15 s and 60 °C for 60 s. The expression levels of various genes were expressed as fold differences in gene expression relative to that of the machined surface.

### 4.6. Mineralization Assay

The mineralization capabilities of cultures were evaluated on day 14 by colorimetry-based assays. The cultures were washed with ddH_2_O and incubated overnight in 1 mL of 0.5 M HCl solution (Sigma-Aldrich) with gentle shaking. The solution was mixed with *o*-cresolphthalein complexone in an alkaline medium (Stanbio LiquidColor, Stanbio, Boerne, TX, USA) to produce a purple calcium–cresolphthalein complexone complex. Color intensity was measured by a multi-detection microplate reader (Synergy^TM^ HT, BioTek Instruments, Inc.) at 550 nm wavelength. The mineralization capability of each titanium disk was also confirmed by visualizing mineralized nodule areas via Alizarin red staining. On day 14 of culture, the specimens were washed twice with 1× PBS at 37 °C and stained for 5 min using 1% Alizarin red (pH 6.3–6.4). The titanium disks were then rinsed twice with ddH_2_O and air-dried.

### 4.7. Cell Detachment Assay

The ability of titanium surfaces to retain cells was evaluated by a chemical detachment assay. At culture days 3, 7, and 14, cells were gently rinsed twice with PBS and treated with 300 μL of 0.05% trypsin–1 mmol/L EDTA–4Na for 1 min at 37 °C. A hematocytometer was used to count the number of detached cells. Then, the remaining cells on the surfaces were completely detached using 300 μL of 0.25% trypsin–1 mmol/L EDTA–4Na for 5 min at 37°C and counted. SEM of the selected cultures was used to confirm the absence of any remaining cells after the second detachment. The percentage of detached cells was calculated according to the following equation: ((Number of detached cells)/(Number of detached cells + Number of remaining cells)) × 100 (%) [18]. Three independent cultures were analyzed for each group, and the data were averaged.

### 4.8. Implant Surgery

Ten-week-old male Sprague–Dawley rats were anesthetized by inhalation of 1%–2% isoflurane. Only left femurs were used to receive an implant. The left leg area was shaved and scrubbed with 10% povidone–iodine solution. The distal aspects of the left femurs were carefully exposed through skin incision and muscle dissection. The flat surface closest to the distal end was selected for implant placement. A 1 mm diameter × 2 mm length implant site was prepared 10 mm from the distal edge of the femur by drilling with a 0.8 mm round burr and was enlarged using reamers (#ISO 090 and 100) as described previously [44,53,56]. For cooling and cleaning, the site was profusely irrigated with a sterile isotonic saline solution. One cylindrical machined, micro-rough, or sub-micro-rough titanium implant was inserted into each prepared hole per femur. Surgical sites were then closed in layers. Muscle and skin were sutured separately with resorbable suture thread. The total number of animals used was 15, distributed among the machined, micro-rough, and sub-micro-rough implant groups. All experiments were performed following a protocol approved by The Chancellor’s Animal Research Committee at the University of California at Los Angeles (ARC #2005-175-41E, approved on 30 January 2018) and followed the PHS Policy for the Humane Care and Use of Laboratory Animals and the UCLA Animal Care and Use Training Manual guidelines.

### 4.9. Biomechanical Implant Push-In Test

The biomechanical implant push-in test was conducted to assess the biomechanical strength of bone–implant integration. The procedure details and method validation are described elsewhere [57,58]. Femurs containing the cylindrical implant were harvested after 2 weeks of healing and embedded into an auto-polymerizing resin with the top surface of the implant horizontal. A testing machine (Instron 5544 electromechanical testing system; Instron, Norwood, MA, USA) equipped with a 2000-N load cell and a pushing rod (diameter, 0.8 mm) was used to load the implant vertically downward at a crosshead speed of 1 mm/min. The push-in value was determined by measuring the peak of the load–displacement curve. 

### 4.10. Statistical Analysis

Data on surface roughness parameters were collected from six sites on three different disks (*n* = 6). All culture studies were performed in triplicate (*n* = 3). Fifteen animals were used for the biomechanical push-in test (*n* = 5). Statistical analyses were carried out using the SPSS (Version 22.0; SPSS Inc., Chicago, IL, USA) statistics program. All statistical analyses were conducted using one-way analysis of variance (ANOVA) followed by a Tukey’s post-test to assess differences among groups. All data are expressed as the group mean ± standard deviation. Results with a probability level of 0.05 or less were considered significant.

## 5. Conclusions

Osteoblastic differentiation was promoted on a newly introduced sub-micro-rough titanium. The effect was even greater than that on micro-rough titanium. The number of attached cells and the cell density were also greater on the sub-micro-rough surface than on the micro-rough surface. However, the sub-micro-rough surface’s in vivo bone integration capability was markedly weaker than that of the micro-rough surface. Thus, the present study reveals disproportionate effects of titanium surface roughness on osteoblastic differentiation and bone integration capability. A new strategy for designing titanium surfaces with better osteoconductivity and minimal interlocking to bone is provided.

## Figures and Tables

**Figure 1 ijms-20-04027-f001:**
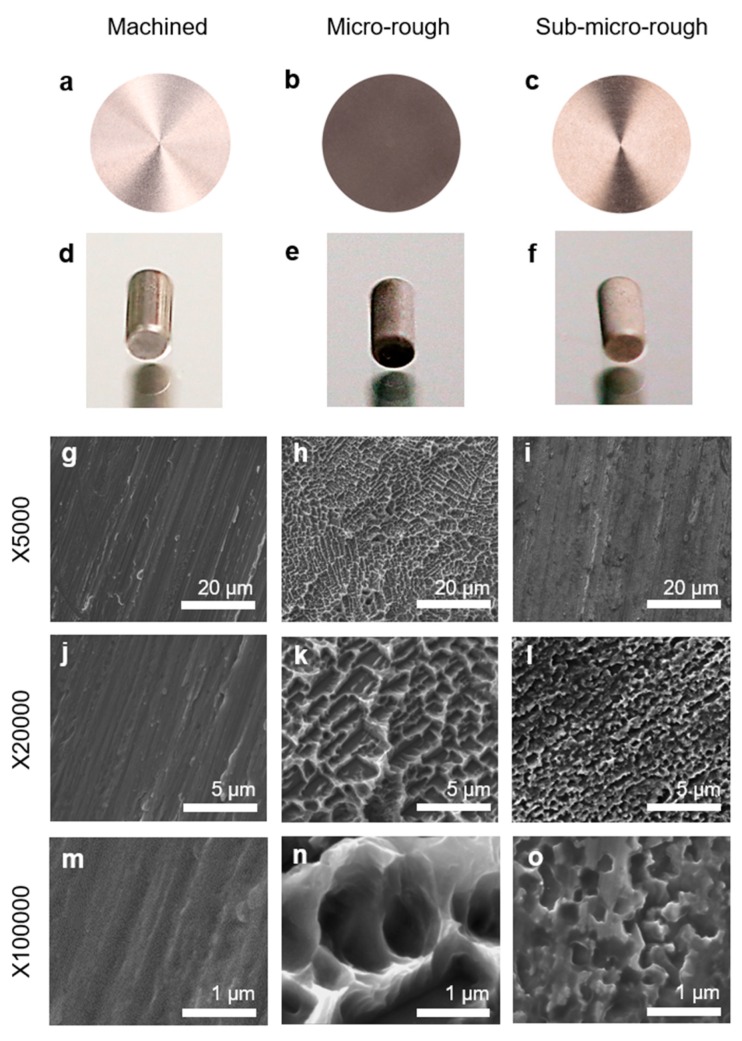
Surface morphology of the titanium disks and implants used in this study. (**a**,**d**) Macroscopic findings of the smooth surface. (**b**,**e**) Macroscopic findings of the micro-rough surface. (**c**,**f**) Macroscopic findings of the sub-micro-rough surface. Scanning electron microscopy photographs showing the surface roughness of the smooth (**g**,**j**,**m**), micro-rough (**h**,**k**,**n**), and sub-micro-rough (**i**,**l**,**o**) surfaces at magnifications of 5000× (**g**,**h**,**i**), 20,000× (**j**,**k**,**l**), and 100,000× (**m**,**n**,**o**). Scale bars = (**g**,**h**,**i**) 20 µm, (**j**,**k**,**l**) 5 µm, and (**m**,**n**,**o**) 1 µm.

**Figure 2 ijms-20-04027-f002:**
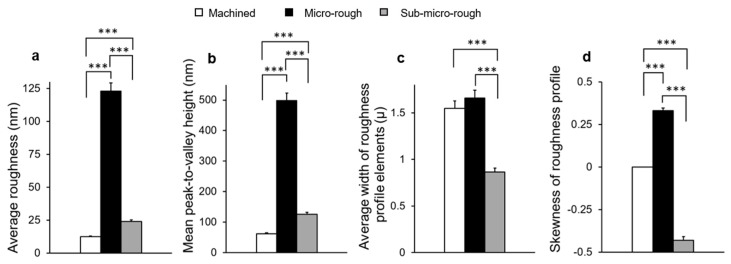
Quantitative measurements of surface roughness of the machined, micro-rough, and sub-micro-rough surfaces using a 3-dimensional profiler. (**a**) Average roughness, (**b**) mean peak-to-valley height, (**c**) average width of roughness profile elements, and (**d**) skewness of roughness profile. Each value represents the mean ± standard deviation of six sites on the three different surfaces (*n* = 6). *** *p* < 0.001, one-way ANOVA followed by Tukey’s test.

**Figure 3 ijms-20-04027-f003:**
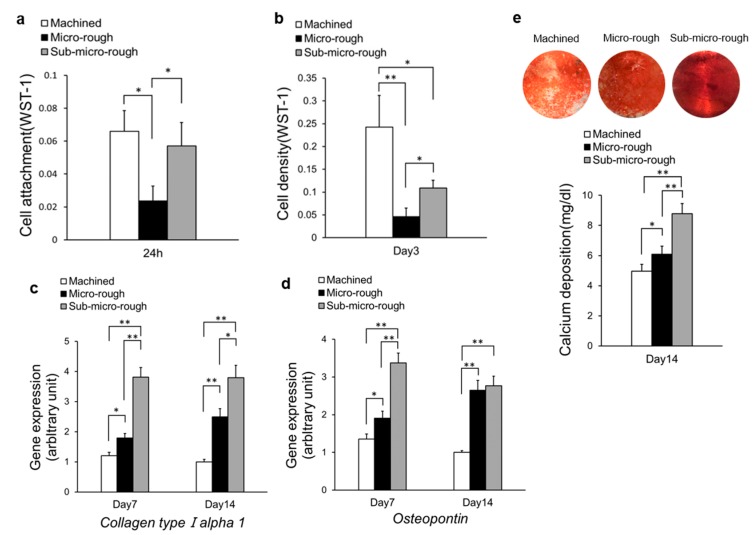
Biological characteristics of osteoblasts on three different titanium surfaces. (**a**) Number of cells attached to each titanium surface during a 24 h incubation, evaluated by a WST-1 assay. (**b**) Cell density evaluated at culture day 3 using a WST-1 assay. (**c**,**d**) Real-time qPCR analysis of mRNA expression of bone-related genes collagen type I alpha 1 and osteopontin on titanium surfaces at day 7 and day 14 using osteoblastic cell cultures. Relative expression levels (2^–ΔΔCt^ values) of the genes of interest were normalized to that of the housekeeping gene *Gapdh*. (**e**) Representative images of mineral deposition evaluated by Alizarin red staining at culture day 14 (top). Colorimetric detection of total calcium deposition measured on the same day (bottom). Each value represents the mean ± standard deviation of triplicate experiments (*n* = 3). * *p* < 0.05, ** *p* < 0.01, one-way ANOVA followed by Tukey’s test.

**Figure 4 ijms-20-04027-f004:**
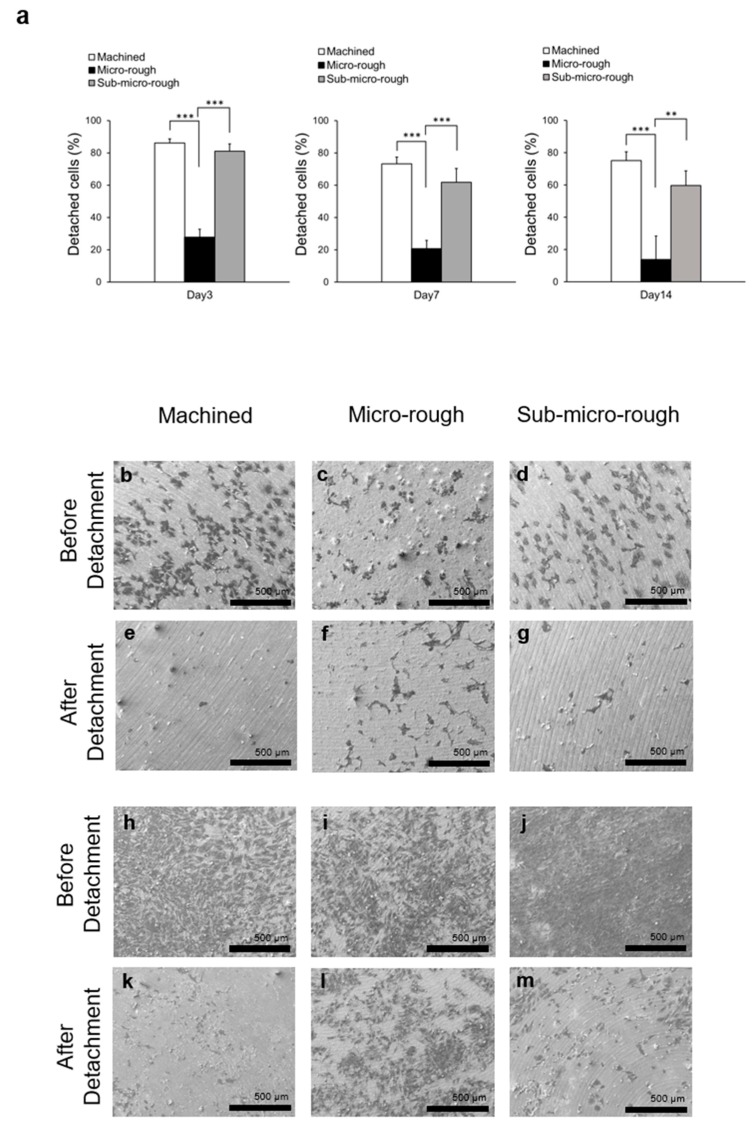
Results of detachment assay on three different titanium surfaces. (**a**) The percentage of detached osteoblasts from three different titanium surfaces after incubation for 3, 7, and 14 days. Each value represents the mean ± standard deviation of triplicate experiments (*n* = 3). ** *p* < 0.01, *** *p* < 0.001, one-way ANOVA followed by Tukey’s test. (**b**–**d**) Representative scanning electron microscopy (SEM) photographs of osteoblasts before detachment from three different titanium surfaces on day 3 of culture. (**e**–**g**) Representative SEM photographs of osteoblasts after detachment from three different titanium surfaces on day 3 of culture. (**h**–**j**) Representative SEM photographs of osteoblasts before detachment from three different titanium surfaces on day 14 of culture. (**k**–**m**) Representative SEM photographs of osteoblasts after detachment from three different titanium surfaces on day 14 of culture. Scale bar = 500 µm.

**Figure 5 ijms-20-04027-f005:**
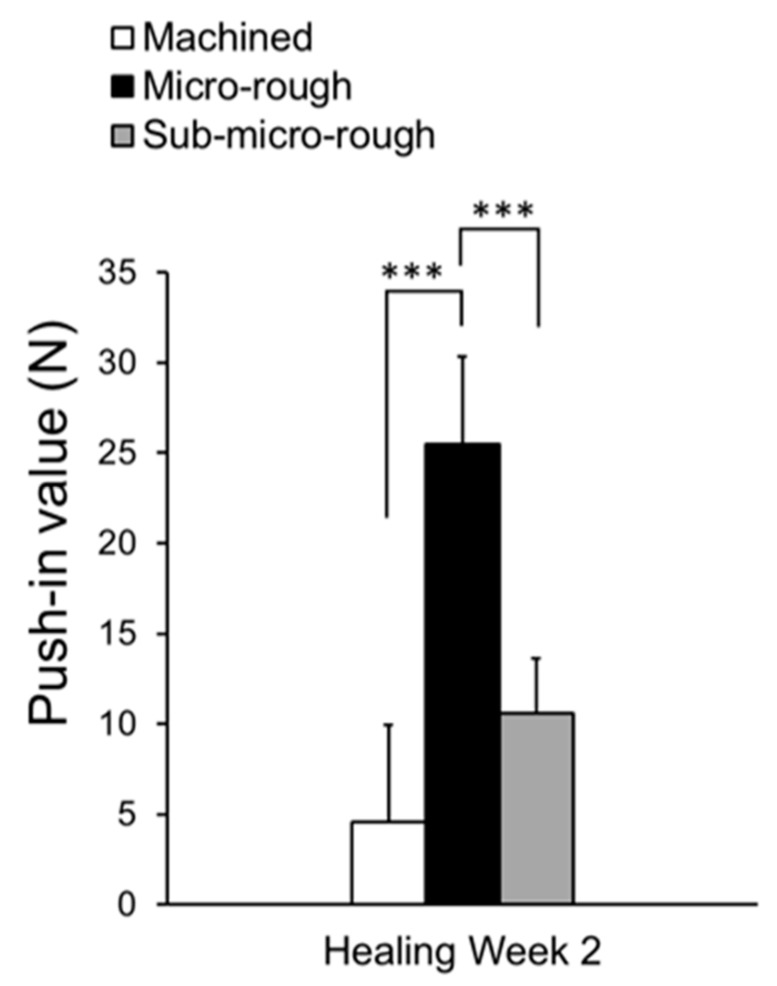
The strength of implant anchorage in bone, evaluated by the biomechanical push-in test in a rat femur model. Each bar represents mean ± standard deviation of machined (*n* = 5), micro-rough (*n* = 5), or sub-micro-rough (*n* = 5) titanium. *** *p* < 0.001, one-way ANOVA followed by Tukey’s test.

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
