# Peer review of "Disproportionate Effect of Sub-Micron Topography on Osteoconductive Capability of Titanium"

_ijms, 2019, doi:10.3390/ijms20164027_

Round 1

Reviewer 1 Report

Very interesting and well-prepared paper.  Three titanium implants with different topographies were prepared to compare theirs in vitro cell interaction, physical and mechanical properties, as well as in vivo integrity. I only have a couple of concerns:

What the implants used for in vivo experiments look like? Maybe a photo of the implant is helpful to clarify the animal procedure.

In the result section, a sentence to explain the reason for performing each section is highly recommended. For example, why tested the cell detachment property? 

Author Response

Manuscript ID: ijms-552923

Title: Disproportionate effect of sub-micron topography on osteoconductive capability of titanium

Response to Reviewer 1 Comments

Point 1: What the implants used for in vivo experiments look like? Maybe a photo of the implant is helpful to clarify the animal procedure.

Thank you very much for your detailed review and suggestions.

Response 1

According to Reviewer 1’s suggestion, we have added images of the implants used in the in vivo experiments (Figure 1d–f). Due to this change and the addition of Figure 1d–f, The Figure 1 legend, Results 2.1. Surface morphology of titanium, and Materials and Methods 4.1. Titanium sample preparation have also been modified.

Point 2: In the result section, a sentence to explain the reason for performing each section is highly recommended. For example, why tested the cell detachment property?

Response 2

→ According to Reviewer 1’s suggestion, we have corrected the text to explain the reason for performing each part of the experiment. The added sentences are as follows:

2.3. “The effect of cell attachment onto titanium surfaces was assessed based on the numbers of cells attached to the titanium surfaces after a 24-h culture period.”

“The number of propagated cells to titanium surfaces was assessed based on the cell density to the titanium surfaces after day three of the culture period.”

“To assess osteoblastic phenotypes on titanium surfaces, bone-related gene analysis was performed using quantitative PCR at two time points.”

“The level of calcium deposition was evaluated as a later-stage osteoblastic phenotype.”

2.4. “The detachment test was performed to determine the cell–titanium surface attachment strength in vitro.”

2.5. “A push-in value that directly represents the shear strength of integration between the bone and implant was used to evaluate the osseointegration.”

Reviewer 2 Report

ijms-552923  

The authors dealt with disproportionate effect of sub-micron topography on osteoconductive capability of titanium. However, there are several criticisms. Suggestions and recommendations are as follows.

It is well known that surface roughness affect osteoblast and improve osseointegration, so what is the novelty of this study? And why did this study choose this surface roughness (is there any preceding study?).

The micro rough was higher than sub micro rough at push in test at 14 day, opposite to the result of gene expression and calcium deposition. Is there histological image that show the aspects of osseointegration?. To clarify the results, author should be appeared the histological images of sample after push-out testing.

L 205: sub micro rough is there any preceding study about the relationship between surface roughness and the mechanical interfacial engagement of cells and bone tissue?

Author Response

Manuscript ID: ijms-552923

Title: Disproportionate effect of sub-micron topography on osteoconductive capability of titanium

Response to Reviewer 2 Comments

Point 1: It is well known that surface roughness affect osteoblast and improve osseointegration, so what is the novelty of this study? And why did this study choose this surface roughness (is there any preceding study?).

Thank you very much for your detailed review and suggestions.

Response 1

→ The novelty of this study is that we developed a sub-micro rough titanium surface that promoted the function of osteoblasts and showed minimal osseointegration.

We performed this study because there is no report about the biological effects of sub-micro rough surfaces in vitro or in vivo.

According to Reviewer 2’s suggestion, we have added sections of text to emphasize the first report and to explain the novelty (See Abstract Lines 14 and 26–28 and Discussion Lines 298–301).

Point 2: The micro rough was higher than sub micro rough at push in test at 14 day, opposite to the result of gene expression and calcium deposition. Is there histological image that show the aspects of osseointegration? To clarify the results, author should be appeared the histological images of sample after push-out testing.

Response 2

We could not show histological images, because we did not take samples after the push-in test at 14 days. However, according to Reviewer 2’s suggestion, we have added the following text regarding the histological consideration after the push-in test (See Discussion Lines 278–297):

“In the present study, we did not histologically evaluate the bone–implant interface of the three kinds of implant after the push-in test. We placed acid-etched implants and machined implants in a rat femur model in a previous study and examined their bone histological imaging around implants with Goldner's trichrome stain [45]. From these results, the bone–implant contact was significantly higher on the acid-etched surface than on the machined surface. Furthermore, we also reported the surface of the acid-etched implant and machined implant after push-in tests with implant surface morphology based on SEM and elemental composition analysis of the implant surface using energy dispersive X-ray spectroscopy (EDS) [52]. The machined implant after a push-in test at four weeks showed exposure without notable bone-like structures, with the accompanying EDS spectrum representing clear and high peaks of titanium and lower peaks of phosphorous and calcium. On the other hand, acid-etched implants showed the robust formation of bone-like structures, which were represented on the EDS spectrum by clear peaks of calcium and phosphorus. Moreover, the sub-micro-rough surface may have a bone–implant contact equivalent to that of the machined surface because there was no significant difference between them in terms of push-in value after two weeks of culture. Considering the above, it may be predicted that many bone-like structures are detached from the implant interface on the sub-micro-rough surface, as on the machined surface. Therefore, we will need to conduct morphological and histological evaluations on the sub-micro-rough surface between implants and bone tissue after push-in tests using histological analysis and SEM in future studies. We should also evaluate the elemental composition of the sub-micro-rough implant surface using EDS.”

Point 3: L 205: sub micro rough is there any preceding study about the relationship between surface roughness and the mechanical interfacial engagement of cells and bone tissue?

Response 3

→ This is the first study on the relationship between the surface roughness and the mechanical interfacial engagement of the cells and bone tissue on the sub-micro-rough surface.

In the same manner as that in Response 1, we have added sections of text to emphasize that this is the first report, as well as to explain the novelty (See Abstract Lines 14 and 26–28 and Discussion Lines 298–301).

Reviewer 3 Report

In ‘Disproportionate effect of sub-micron topography on osteoconductive capability of titanium’, Saruta et al. compared titanium surfaces with different roughness, looking at cell attachment and proliferation, calcium deposition and expression of osteogenic genes, and bone-to-implant integration. Different surfaces performed better in the different assays performed.

In general, the study is clearly reported; however, my concern is about the type of cells that were used. Throughout these are called osteoblasts or osteoblastic cells, but yet osteoblastic differentiation is being measured. It is unclear to me how cells that are already osteoblasts would be expected to undergo further osteoblastic differentiation. Additionally, these cells seem to have been taken from the bone marrow, so perhaps instead they are some sort of progenitor or stem cell?

Additional specific comments:

Abstract: It would be helpful to report what kind of cells were used in the in vitro tests and also what animal model was used for the in vivo evaluation in the abstract.

Abstract: It is unclear how the different surfaces were categorized. In particular, microscale topographies are defined as 1-5 microns, but the surface that was classified as micro-rough had an average roughness of 123 nm.

Line 83: Is this also reporting a peak to peak distance?

Figure 2: Is the average width of roughness profile elements in microns? Does this metric have any relation to the peak-to-peak distances measured in the SEM images?

Section 2.3: What type of osteoblasts are being used in these experiments (Line 120)? Later in this section, they are evaluated for expression of Col1a1 as an early marker of osteoblastic differentiation (Lines 130-131). Are the cells differentiated or not at the start? Likewise, what is the relevance of osteopontin, as a mid- to late-stage osteoblastic marker (Lines 137-138)?

Section 2.3: What is the cell density on days 7 and 14 when the gene expression was analyzed?

Figure 4: Are the SEM images showing the samples after the first detachment step or after the second, as described in the methods (Lines 328-329)?

Discussion: The authors raise the point about the balance between proliferation and differentiation. In this study, it seems that only a differentiation medium has been used (with ascorbic acid, beta-glycerophosphate, and dexamethasone). How does this influence the ability of the cells to proliferation vs. differentiate? Some discussion of this should be added in the manuscript.

Line 272: What are bone marrow-derived osteoblastic cells? Are these mesenchymal stem cells? Have they been pre-differentiated to become osteoblasts?

Author Response

Manuscript ID: ijms-552923

Title: Disproportionate effect of sub-micron topography on osteoconductive capability of titanium

Response to Reviewer 3 Comments

Point 1: In general, the study is clearly reported; however, my concern is about the type of cells that were used. Throughout these are called osteoblasts or osteoblastic cells, but yet osteoblastic differentiation is being measured. It is unclear to me how cells that are already osteoblasts would be expected to undergo further osteoblastic differentiation. Additionally, these cells seem to have been taken from the bone marrow, so perhaps instead they are some sort of progenitor or stem cell?

Thank you very much for your detailed review and suggestions.

Response 1

According to Reviewer 3’s suggestion, we have added and corrected the following section explaining osteoblasts (See Discussion Lines 222-242).

“We examined the cell attachment, proliferation, and functional phenotypes of osteoblast-like rat bone marrow in this study. The rat bone marrow cells used in this experiment are known to differentiate towards an osteoblast-like phenotype when supplemented with dexamethasone and beta-glycerophosphate [21-24]. In addition, many studies related to osteoblasts have been reported using various cells, such as bone-marrow-derived mesenchymal stem cells, MC3T3-E1, and MG-63 [25-31]. Osteoblast differentiation methods have been reported to use many kinds of reagents, including 1,25-dihydroxyvitamin D3 [32], hormones [33,34], growth factors [35,36], bone morphogenetic proteins [37,38], aluminum chloride (AlCl3) [39], sodium fluoride (NaF) [35], prostaglandins [40,41], β-glycerophosphate [30], and ascorbic acid [31]. There are also many reports on the timing of administering differentiation-inducing reagents to these cells [27,42,43]. Cells harvested from rat bone marrow were differentiated in this study using osteogenic induction medium containing ascorbic acid, β-glycerophosphate, and dexamethasone. Since titanium materials do not have osteogenic induction capability in the material itself, it was necessary to differentiate and induce cells before seeding in order to examine osteoconduction on each type of titanium surface in the present study. In past studies, we have reported many changes in the differentiation, proliferation, and genetic phenotype of mature osteoblast cells before seeding cells using the osteogenic induction medium [44-47]. It is necessary to use various osteogenic induction reagents, change the timing of the addition of the osteogenic induction medium, and conduct experiments in a comprehensive manner in future studies. Moreover, since cells harvested from bone marrow cells did not use cell-sorting in this study, we should also conduct detailed studies on sub-micro-rough surfaces using rat and human mesenchymal stem cells.”

Point 2: Abstract: It would be helpful to report what kind of cells were used in the in vitro tests and also what animal model was used for the in vivo evaluation in the abstract.

Response 2

→ In accordance with Reviewer 3’s suggestion, we have added this information into the Abstract (Lines 18–22):

Lines 18–20: “In culture studies using bone marrow-derived osteoblasts, the micro-rough surface showed the lowest proliferation and fewest cells attaching during the initial stage.”

Lines 21–22: “The bone–implant integration in the Sprague–Dawley male rat femur model was the strongest on the micro-rough surface.”

Point 3: Abstract: It is unclear how the different surfaces were categorized. In particular, microscale topographies are defined as 1-5 microns, but the surface that was classified as micro-rough had an average roughness of 123 nm.

Response 3

The micro- (1–5 μm) and sub-micro (0.1–0.5 μm)-scale topographies indicate compartmental structures determined from SEM images. The calculated average roughness values of 12.5 ± 0.65 nm (machined), 123 ± 6.15 nm (micro-rough), and 24 ± 1.2 nm (sub-micro-rough) are shown in Figure 2a. We have corrected the following text in the Abstract (Lines 14–18), because the explanation of the method was unclear.

Line 14–18: “We compared osteoblastic phenotypes and in vivo bone and implant integration abilities between titanium surfaces with micro- (1–5 µm) and sub-micro-scale (0.1–0.5 µm) compartmental structures and machined titanium. The calculated average roughness was 12.5 ± 0.65 nm, 123 ± 6.15 nm, and 24 ± 1.2 nm for machined, micro-rough, and sub-micro-rough surfaces, respectively.”

Point 4: Line 83: Is this also reporting a peak to peak distance?

Response 4

Line 83: The sub-micro-scale (pits were 0.1–0.5 μm, with an average size of 0.15 μm) topographies indicate the compartmental structures determined from the SEM images (Figure 1o).

Point 5: Figure 2: Is the average width of roughness profile elements in microns? Does this metric have any relation to the peak-to-peak distances measured in the SEM images?

Response 5

Thank you very much for your valuable feedback.

The average roughness (Ra) is one of the most commonly used height direction parameters. Ra is the average value of the average height difference from the average surface and can be expressed using a formula. Therefore, Ra is a parameter based on peak-to-peak distances measured using the SEM images.

Point 6: Section 2.3: What type of osteoblasts are being used in these experiments (Line 120)? Later in this section, they are evaluated for expression of Col1a1 as an early marker of osteoblastic differentiation (Lines 130-131). Are the cells differentiated or not at the start? Likewise, what is the relevance of osteopontin, as a mid- to late-stage osteoblastic marker (Lines 137-138)?

Response 6

Thank you very much for your detailed review and suggestions.

According to Reviewer 3’s suggestion, we have added and corrected the following text regarding osteoblasts (See Discussion Line 231-242).

“Cells harvested from rat bone marrow were differentiated in this study using osteogenic induction medium containing ascorbic acid, β-glycerophosphate, and dexamethasone. Since titanium materials do not have osteogenic induction capability in the material itself, it was necessary to differentiate and induce cells before seeding in order to examine osteoconduction on each type of titanium surface in the present study. In past studies, we have reported many changes in the differentiation, proliferation, and genetic phenotype of mature osteoblast cells before seeding cells using the osteogenic induction medium [44-47]. It is necessary to use various osteogenic induction reagents, change the timing of the addition of the osteogenic induction medium, and conduct experiments in a comprehensive manner in future studies. Moreover, since cells harvested from bone marrow cells did not use cell-sorting in this study, we should also conduct detailed studies on sub-micro-rough surfaces using rat and human mesenchymal stem cells.”

Point 7: Section 2.3: What is the cell density on days 7 and 14 when the gene expression was analyzed?

Response 7

→ Since the cells used in this experiment and the seeded cell densities become confluent after approximately five days, it is difficult to evaluate the correct cell density among groups. Therefore, the cell densities at days 7 and 14 were not measured when the gene expression was analysed.

Point 8: Figure 4: Are the SEM images showing the samples after the first detachment step or after the second, as described in the methods (Lines 328-329)?

Response 8

→ Figure 4 e–g and k–m showed the SEM images after the first detachment step.

Point 9: Discussion: The authors raise the point about the balance between proliferation and differentiation. In this study, it seems that only a differentiation medium has been used (with ascorbic acid, beta-glycerophosphate, and dexamethasone). How does this influence the ability of the cells to proliferation vs. differentiate? Some discussion of this should be added in the manuscript.

Response 9

Thank you very much for your detailed review and suggestions.

According to Reviewer 3’s suggestion, we have added and corrected the following text to better discuss the balance between proliferation and differentiation osteoblasts (See Discussion Line 265-270).

“However, since we are using an osteogenic induction medium in these experiments, it is possible that osteoblasts are somewhat emphasized in differentiation rather than proliferation. Since titanium materials do not have osteoinduction capabilities, the cells require an osteogenic induction medium. Therefore, we need to further investigate the effects of the surfaces by creating a concentration gradient in the osteogenic induction medium and reducing the reagents contained in it.”

Point 10: Line 272: What are bone marrow-derived osteoblastic cells? Are these mesenchymal stem cells? Have they been pre-differentiated to become osteoblasts?

Response 10

Thank you very much for your detailed review and suggestions.

According to Reviewer 3’s suggestion, we have added and corrected the following text to better explain osteoblasts (See Discussion Line 231-242).

“Cells harvested from rat bone marrow were differentiated in this study using osteogenic induction medium containing ascorbic acid, β-glycerophosphate, and dexamethasone. Since titanium materials do not have osteogenic induction capability in the material itself, it was necessary to differentiate and induce cells before seeding in order to examine osteoconduction on each type of titanium surface in the present study. In past studies, we have reported many changes in the differentiation, proliferation, and genetic phenotype of mature osteoblast cells before seeding cells using the osteogenic induction medium [44-47]. It is necessary to use various osteogenic induction reagents, change the timing of the addition of the osteogenic induction medium, and conduct experiments in a comprehensive manner in future studies. Moreover, since cells harvested from bone marrow cells did not use cell-sorting in this study, we should also conduct detailed studies on sub-micro-rough surfaces using rat and human mesenchymal stem cells.”

Round 2

Reviewer 3 Report

Overall, the authors have improved their manuscript and addressed most of my comments. I only have two small points that should be addressed before publication:

Lines 262 and 417: Is the proliferation rate (would need to calculate this relative to day 1) changing or the cell density (as stated in the results) different among the different groups?

Line 292: Should this be 2 weeks of implantation instead of 2 weeks of culture?

Author Response

Manuscript ID: ijms-552923

Title: Disproportionate effect of sub-micron topography on osteoconductive capability of titanium

Responses to Reviewer 3’s Comments

Point 1: Lines 262 and 417: Is the proliferation rate (would need to calculate this relative to day 1) changing or the cell density (as stated in the results) different among the different groups?

Response 1

Thank you very much for your detailed review and suggestions.

We cannot provide data on the proliferation rate, as we did not directly measure cell proliferation in this study. Furthermore, we did not directly measure cell numbers; rather, we determined cell density based on the value of WST-1 on day 3 in the different groups. Therefore, according to Reviewer 3’s suggestion, we have corrected the following text (see lines 261 and 416):

Line 261 “Given this, we anticipated that the sub-micro-rough surface would show a higher cell density than the micro-rough surface, which turned out to be true.”

Line 416 “The number of attached cells and the cell density were also greater on the sub-micro-rough surface than on the micro-rough surface.”

Point 2: Line 292: Should this be 2 weeks of implantation instead of 2 weeks of culture?

Response 2

→ In accordance with Reviewer 3’s suggestion, we have corrected the following text (see Discussion, line 291):

“Moreover, the sub-micro-rough surface may have a bone–implant contact equivalent to that of the machined surface because there was no significant difference between them in terms of push-in value after 2 weeks of implantation.”